# Identification and Characterization of a New Type III Polyketide Synthase from a Marine Yeast, *Naganishia uzbekistanensis*

**DOI:** 10.3390/md18120637

**Published:** 2020-12-11

**Authors:** Laure Martinelli, Vanessa Redou, Bastien Cochereau, Ludovic Delage, Nolwenn Hymery, Elisabeth Poirier, Christophe Le Meur, Gaetan Le Foch, Lionel Cladiere, Mohamed Mehiri, Nathalie Demont-Caulet, Laurence Meslet-Cladiere

**Affiliations:** 1Laboratoire Universitaire de Biodiversité et Ecologie Microbienne, University Brest, F-29280 Plouzané, France; laure.martinelli@univ-st-etienne.fr (L.M.); vanessa.redou@gmail.com (V.R.); bastien.cochereau@etu.univ-nantes.fr (B.C.); nolwenn.hymery@univ-brest.fr (N.H.); Elisabeth.Poirier@univ-brest.fr (E.P.); christophe.lemeur@univ-brest.fr (C.L.M.); Gaetan.lefloch@univ-brest.fr (G.L.F.); 2Integrative Biology of Marine Models (LBI2M), Station Biologique de Roscoff (SBR),CNRS, UMR8227, Sorbonne Université, 29680 Roscoff, France; delage@sb-roscoff.fr (L.D.); cladiere@sb-roscoff.fr (L.C.); 3Marine Natural Products Team, CNRS, UMR 7272, Institut de Chimie de Nice, Université Côte d’Azur, 06108 Nice, France; mohamed.mehiri@unice.fr; 4UMR ECOSYS, INRAE, INRAE, University of Paris, 78026 Versailles, France; nathalie.demont-caulet@inra.fr; 5AgroParisTech, Université Paris-Saclay, 78026 Versailles, France

**Keywords:** marine yeast, PKSIII, triketide pyrones, pentaketide resorcinols, cytotoxic activity

## Abstract

A putative Type III Polyketide synthase (PKSIII) encoding gene was identified from a marine yeast, *Naganishia uzbekistanensis* strain Mo29 (UBOCC-A-208024) (formerly named as *Cryptococcus* sp.) isolated from deep-sea hydrothermal vents. This gene is part of a distinct phylogenetic branch compared to all known terrestrial fungal sequences. This new gene encodes a C-terminus extension of 74 amino acids compared to other known PKSIII proteins like *Neurospora crassa*. Full-length and reduced versions of this PKSIII were successfully cloned and overexpressed in a bacterial host, *Escherichia coli* BL21 (DE3). Both proteins showed the same activity, suggesting that additional amino acid residues at the C-terminus are probably not required for biochemical functions. We demonstrated by LC-ESI-MS/MS that these two recombinant PKSIII proteins could only produce tri- and tetraketide pyrones and alkylresorcinols using only long fatty acid chain from C8 to C16 acyl-CoAs as starter units, in presence of malonyl-CoA. In addition, we showed that some of these molecules exhibit cytotoxic activities against several cancer cell lines.

## 1. Introduction

Secondary metabolites represent an attractive and important source of natural products with a wide range of bioactivities and biotechnological applications. Organisms such as plants, algae, bacteria and fungi are able to produce bioactive compounds, many of which have been used as antitumor or antimicrobial agents, immunosuppressants, vaccine antigens or insecticides. Fungi especially are known to be one of the major producers of bioactive compounds. In fact, in 2012 the number of bioactive natural compounds characterized was 80,000–100,000, of which, 15,000 were of fungal origin [1]. Some well-known examples of bioactive fungal compounds include penicillin, developed as an antibacterial [2], lovastatin developed as a cholesterol-lowering medication [3,4], cyclosporins developed as immunosuppressive agents [5] and taxol, which has been proved to have an antitumor activity [6,7]. Up to now, most of the characterized fungal natural products are from terrestrial habitats even though fungi also colonize fresh water and marine habitats. Aquatic fungi have been less well studied compared to terrestrial fungi, but they have generated an increasing level of interest in the last decade.

Fungi from marine habitats were first studied in the 1940s when Barghoorn and Linder (1944) demonstrated their occurrence [8]. First exhaustive definition of marine fungi distinguished obligate and facultative organisms. In other words, marine fungi were divided between those “that grow and sporulate exclusively in a marine or estuarine habitat from those from freshwater or terrestrial milieus able to grow and possibly to sporulate in the marine environment” [9]. In 2015, Jones et al. provided a review of the classification of marine fungi, including the accepted names of 1112 species of characterized fungi from marine habitats [10]. Recently, there has been an increased level of interest in marine fungi, especially in defining their diversity and ecological role. Along with studies on marine filamentous fungi and yeasts, several studies have focused on the diversity of fungi in extreme environments such as the subseafloor [11,12], hydrothermal fields [13], or deep-sea hydrothermal vents [14,15]. In many cases, their activity and function has also been described, using transcriptomics to better understand their ecological role in extreme ecosystems [16]. Such studies have suggested that fungal communities from extreme ecosystems might represent an untapped reservoir of potential novel bioactive molecules, and, in recent years, secondary metabolites (also called specialized metabolites) obtained from fungi isolated from fresh water and marine habitats have gained considerable attention. According to a study by Rateb and Ebel (2011), marine fungal secondary metabolites have various biological and pharmaceuticals properties. In total, 690 novel structures of secondary metabolites have been identified from marine fungi between mid-2006 and 2010 [17]. At this time, 6222 molecules have been referenced from Dikarya fungi in MarineLit database (http://pubs.rsc.org/marinlit/). Evaluation of the biotechnological potential of marine-derived fungi has revealed that they produce mainly polyketides, terpenoids and peptides [17,18]. In addition, whole genome sequence analysis suggests that many deep-sea fungi have the potential to produce bioactive compounds. Indeed, Rédou et al. (2015) provided evidence that almost all fungal isolates in their study (96% of the 200 filamentous fungi and yeasts) had at least one gene involved in a secondary metabolite biosynthesis pathway [19], for example genes that encode enzymes involved in polyketides production (Polyketide Synthase, PKS), terpenoids (Terpene synthases, TPS) or non-ribosomal peptides (Non-Ribosomal Peptide Synthetase, NRPS). Along with this study, several studies have demonstrated antimicrobial activities of diverse marine fungi [20], deep-subseafloor fungi [21], and antimicrobial, antitumoral and antioxidant activities of sponge-associated marine fungi [22].

Polyketides represent a major and highly diverse group of natural products [17,23]. With a wide range of complex structures including macrolides, polyphenols, polyenes, and polyethers, polyketides comprise several groups of biologically important secondary metabolites such as flavonoids, phloroglucinols, resorcinols, stilbenes, pyrones, curcuminoids and chalcones [24,25]. Polyketides have numerous functions and applications as pigments, antibiotics, immunosuppressants, antioxidants, antiparasitics, cholesterol-lowering, and antitumoral agents [26]. Polyketide biosynthesis requires specific enzymes known as polyketide synthases (PKS). PKSs are a large group of enzymes, divided into three classes: type I, II or III [26,27], where type I PKSs are large multi-domain enzymes able to function in either a modular or iterative manner, type II PKSs are dissociable multi-enzyme complexes functioning in an iterative manner and type III PKSs are homodimeric enzymes, mechanistically different from the two other subgroups of PKSs, and structurally simpler [25,28]. Type III PKSs share common characteristics: they form dimers. Each monomer with a size of 40–45 kDa contains a conserved Cys-His-Asn catalytic triad within an active site cavity [25]. Polyketides are produced by iterative decarboxylative condensations of malonyl-CoA and diverse acyl-CoA thioesters (as extender and starter units), and cyclization reactions [28]. Despite their simplicity type III PKSs have an unusually wide range of substrates and play an important role in the biosynthesis of various bioactive polyketide compounds in different organisms [29]. For example, chalcones are produced by land plants [30,31]. Phlorotannins are unique to brown algae and phloroglucinols can be synthesized by diverse kind of organisms from prokaryotes (*Pseudomonas*) to eukaryotes (brown algae) [31,32,33,34,35]. Type III PKSs have also recently been discovered in fungi [36]. The first characterized fungal type III PKS was isolated from *Neurospora crassa* [37] and since then, several fungal type III PKSs have been reported. Among them, type III PKSs isolated from *Aspergillus oryzae* [38,39,40] and another from *Aspergillus niger* [41] have been characterized, as well as a type III PKS isolated from *Sporotrichum laxum* which appears to be involved in the production of spirolaxine, a potential drug candidate with anti-*Helicobacter pylori* activity [42]. A recent analysis reveals a total of 552 *type III pks* genes from 1193 fungal genomes (JGI Mycocosm) [43]. Only eleven type III PKSs have been biochemically characterized in fungal kingdom [43]. Polyketides produced by fungal type III PKS are grouped into triketide pyrones, tetraketide pyrones and alkylesorcinols [25]. All enzymes are isolated from terrestrial fungi. To date, no marine fungal enzymes have been described.

In this study we report the identification of a type III PKS from a marine yeast *Naganishia uzbekistanensis* strain Mo29 (UBOCC-A-208024) (formerly named as *Cryptococcus* sp. [44]) [14,45]. Two enzyme forms, the whole protein PKSIII Mo29 (Std) and the reduced protein PKSIII Mo29 (Red) without the C-terminus extension were expressed as recombinant proteins in *E. coli* and their biochemical enzyme activities were analyzed by LC/MS-MS. Our phylogenetic, biochemical and structural modelling of this new fungal type III PKS provide new insights into the biotechnological potential of deep-sea yeast.

## 2. Results

### 2.1. Identification and Phylogenetic Analysis of Pksiii Gene from Marine Yeast

We have previously sequenced the genome of *N. uzbekistanensis* strain Mo29 (UBOCC-A-208024) [45] and identified a putative *pksIII* gene using the AntiSMASH v5 tool [46]. The amino acid sequence was submitted to GenBank with the accession number MW324483.

This analysis revealed that the putative encoded protein had 38.84% similarity and 32.70% identity to PKSIII of *N. crassa* ATCC 24698 (PKSIIINc) [47]. However, the putative PKSIII is 536 aa in length compared to 465 aa in *N. crassa* PKSIII protein, a C-terminus extension with 74 aa. A secondary structure for PKSIII Mo29 is predicted with Phyre2 [48] (Appendix A).

A phylogenetic analysis was performed on 755 representative PKSIII proteins from eukaryotes and bacteria, including functionally characterized enzymes known to accept a diversity of CoA thioester starter units and including the 552 genes discovered in the study of Navarro-Munoz and Collemare [43]. The maximum likelihood phylogenetic analysis of 755 PKS III sequences led to the distinction of three major branches (Figure 1). Each one contains sequences of a specific phylum corresponding to fungi, bacteria and plants, respectively. Bacterial and fungal sequences are separated from plants at a well-supported node (bootstrap = 99) suggesting that PKSIII proteins from Chlamydia and firmicutes are more closely related to fungi. In fungi, the only Chytrid sequence is positioned at the base of the tree similar to recent findings by Navarro-Munoz and Collemare [43]. The majority of fungal PKSIII sequences (619) are grouped together but two distinct branches of basidiomycetes are present. The closest is composed of 37 sequences and is supported by a strong bootstrap value of 100. The other is represented by four sequences: the PKSIII from *N. uzbekistanensis* strain Mo29 (UBOCC-A-208024), *Naganishia vishniacii*, *Cryptococcus wieringae* and *Phaffia rhodozyma*. This result shows that the separation of the common ancestor of the *N. uzbekistanensis* PKSIII and the major part of all fungal sequences protein is relatively ancient in the evolution of fungal PKSIII. 

Furthermore, the fact that few phylogenetic relatives are present in the same clade might mean that this is a rare sequence encountered in nature. This latter result has to be taken with caution because the available databases are not fully completed and in particular novel marine fungi could be sequenced in the future. Interestingly, all the four sequences possess a C-terminal extension. This C-terminus extension 74 amino acids of these PKSIII have no similarity to any known protein. We also sought to determine whether these distinct PKSIII proteins could be associated with organisms from a specific habitat; however, further analysis of the origin of the isolates revealed that *Cryptococcus wieringae* (*Filobasidium wieringae*) and *Phaffia rhodozyma* (*Xanthophyllomyces dendrorhous*) were collected in temperate terrestrial forests (NCBI biosample information and JGI mycocosm ressources), *N. vishniacii* (*Naganishia vishniacii*) is present in Antarctica and *N. uzbekistanensis* strain Mo29 (UBOCC-A-208024) was isolated from hydrothermal vent. Therefore, no correlation of similar niches could be attributed between the species encoding this distinct type of PKSIII protein.

### 2.2. Identification of the Key Molecular Features by Structural Modelling of N. Uzbekistanensis PKSIII Protein

We analysed the putative structure of the PKSIII enzyme from *N. uzbekistanensis* using SWISS MODEL [49]. The structure of the type III PKS enzyme from *N. crassa* (PKSIIINc) (PDB: 3E1H) which shares 38.84% amino acid identity with PSKIII from Mo29, was used as a template [47]. A model was generated included the C-terminus extension of 74 amino acids of PKSIII Mo29 (Appendix A). However, we were unable to model the C-terminus extension which have no similarity to any known PKSIII structures (Appendix A). The modelling revealed that this protein could form a dimer complex, like other fungal PKSIII (Figure 2A). We compare PKSIII Mo29 amino acids involved in substrate binding with those found in PKSIII from PKSIII from *N. crassa* (PKSIIINc) [37,47,50], *Mycobacterium tuberculosis* (PKS18) [51], two PKSIII from *Aspergillus oryzae* (AoCsyA and AoCsyB) [38,39,52], PKSIII from *Botrytis cinerea* (BcPKS) [53] and PKSIII from *Aspergillus niger* (AnPKS) [41]. The active site residues cysteine 171, histidine 351 and asparagine 385 (catalytic sites) are conserved (Table 1 and Figure 2B) (Appendix A).

In addition, residues involved in the cyclisation of malonyl-CoA were also conserved (cysteine 139 and threonine 140) (Figure 2B) [50]. Previous work has demonstrated that serine at position 186 is essential for condensation of aldols in PKSIIINc [54]. As shown in Figure 2B, in PKSIII Mo29, the amino acid at the equivalent position is also a polar residue—threonine 208 (Table 1 and Appendix A). Our model also revealed the presence of a cysteine at position 139. The equivalent position in PKSIII of *N. crassa* (position 120) is a key residue in determining enzymatic function. It has been shown that substitution of a C120 by a phenylalanine or serine in PKSIIINc results in synthesis of phloroglucinol instead of α-pyrone [50]. The active site is flanked by the acyl-binding tunnel similar to those observed in the structures of bacterial and fungal PKSIII [38,39,47,50,53]. We observed some differences with residues observed in the tunnel of PKSIIINc [47].

The residues in PKSIIINc 125N, 189M, 190V and 206G are substituted by 144Y, 211L, 212C and 236S in PKSIII Mo29 (Table 1 and Figure 2) (Appendix A). When stearoyl-CoA is docked into the substrate binding site of BcPKS, the starter is bound through hydrogen bonds with 319G, 321A and 322T [53]. These residues are substituted by 354G, 355S and 356L in PKSIII Mo29 (Table 1 and Appendix A). Thus, our analysis indicates that PKSIII Mo29 is likely to be a pyrone synthase, similar to PKSIII in terrestrial fungi like *N. crassa*, *B. cinerea* and *A. niger*. In order to confirm this function, we next sought to overexpress the protein in *E. coli*.

### 2.3. Expression and Enzymatic Activities of Recombinant PKSIII Mo29 (Full Length and Without C-Terminus Extension)

To compare to other fungal PKSIII, this protein has 74 aa extension in C-terminus. To understand the function of this extension, two constructs were made. The full length (Std) and the truncated/reduced gene (without the 74 aa extension) (Red) were cloned into pQE 80L plasmid to overexpress the two proteins in *E. coli* (both constructs were overexpressed without 30 aa of the putative N-terminal signal peptide); a previously published strategy used to express proteins from *Sordaryomycetes* fungi and *B. cinerea* [55]. As shown in Figure 3, purified PKSIII Mo29 Std and PKSIII Mo29 Red proteins migrated as 54 and 48 kDa proteins in SDS-PAGE, consistent with the predicted theoretical mass of 54.329 kDa calculated (without 30 aa of the putative N-terminal signal peptide). An additional band was detected in both purified samples with high molecular weight. However, the Dynamic Light Scattering (DLS) analysis showed that a homogenous pure protein was present in the sample. The size of pure protein was also calculated with marker on a Sephacryl-200 gel filtration column. Therefore, our results suggest that PKSIII Mo29 forms a dimeric complex, shown with the model generated previously (Figure 2A) and as already shown for all PKSIII including PKSIIINc [56].

Activities of pyrone synthases of fungi, including *A. niger* [57], in *N. crassa* [47] or *A. oryzae* [52] have been characterized biochemically by LC/MS-MS or GC/MS-MS. These fungi mainly produce triketide pyrones, tetraketide pyrones and pentaketide pyrones. However, it has also been shown in *N. crassa*, that depending on the acyl-CoA used as starter, the production of alkylresorcinols was increased [47].

In vitro tests with both recombinant PKSIII Mo29 (Std and Red) showed that these enzymes could not use short chain molecules as a starter. No product was detected by HPLC-ESI/MS-MS (negative mode) when the starters are acetyl-CoA (C2CoA) and hexanoyl-CoA (C6CoA). The first compound produced by both forms of recombinant PKSIII Mo29 is C_12_H_18_O_3_ (*m*/*z* 209.1300) in the presence of malonyl-CoA (C3CoA) and octanoyl-CoA (C8CoA) as a starter (Table 2), although this compound was only detected at low levels. An α-pyrone is also produced by the two forms of proteins in the presence of malonyl-CoA and decanoyl-CoA as a starter. Its molecular formula is C_14_H_22_O_3_ (*m*/*z* 238.1567). Using several collision energies, the molecule is fragmented and the fragment ion has a *m*/*z* of 194.1599 (Table 2). In the presence of malonyl-CoA and lauroyl-CoA, the molecule detected is a C_16_H_26_O_3_ (*m*/*z* 266.1884). The ion produced after fragmentation has a mass *m*/*z* of 222.1952. The final starter tested was palmitoyl-CoA. The two forms of PKSIII Mo29 produce a molecule of mass *m*/*z* of 322.2507. The formula of this molecule is C_20_H_34_O_3_. This formula is confirmed by the ions generated after fragmentation (*m*/*z* 278.2554) (Table 2). Therefore, In vitro experiments revealed that PKSIII Mo29 is active and produces mainly triketide pyrones.

But, the amounts of these molecules produced in vitro, are not the same according to the protein used. Indeed, PKSIII Mo29 Red seems to produce less product than the PKSIII Mo29 Std (Figure 4). With the same quantities of pure protein and substrates, only half amounts of molecules are produced if we compare results with both recombinant Mo29 proteins. For example, the octanoyl pyrone (C_14_H_22_O_3_) is twice as much produced by the PKSIII Mo29 Std compared to the same compound produced by PKSIII Mo29 Red (Figure 4).

Both forms (PKSIII Std and PKSIII Red) of the enzyme can also produce tretraketide pyrones and alkylresorcinols. Consistent with this, we found multiple molecules of the pyrone and alkylresorcinol type in the *E. coli* culture medium following expression of these proteins, which were absent from the culture medium of *E. coli* containing the empty plasmid. All molecules detected are listed in Table 3, and the formulas were validated by fragmentation of the molecules (Table 3 and Figure 5A). Some molecules appear to be produced at higher concentrations than others. Statistical analysis reveals that the five most abundant molecules are triketide pyrones and alkylresorcinols (Figure 5A,B).

Formulas of these compounds are C_22_H_36_O_3_ (*m*/*z* 348.2663), C_20_H_32_O_3_ (*m*/*z* 320.2349), C_23_H_38_O_2_ (*m*/*z* 346.2680), C_20_H_34_O_3_ (*m*/*z* 322.2507) and C_21_H_34_O_2_ (*m*/*z* 318.2555). As shown in the precedent In vitro experiments, the amount of all molecules is different according to the PKSIII Mo29 considered. PKSIII Mo29 Std seems to produce more amount molecules than PKSIII Mo29 Red (Figure 5B). Other molecules found in the supernatant of *E. coli* are detected and characterized but are less expressed by both forms of PKSIII Mo29 than the 5 most abundant molecules (Table 3 and Figure 5B).

### 2.4. Cytotoxicity Activity against Tumoral Cell Lines

Several α-pyrones and alkylresorcinols produced by marine and terrestrial fungi have shown cytotoxic activities against several cancer cell lines [58,59,60,61]. Therefore, molecules produced by PKSIII Mo29 could have cytotoxic effects on human tumoral cell lines. Different models were used to evaluate cell viability. Cells were exposed with different molecules produced in vitro by PKSIII Mo29 (molecules obtained previously) (All molecules are concentrated at 0.012 g/L). After 48 h of exposure to the compound with a molecular formula of C_20_H_34_O_3_ (*m*/*z* 321.2400) produced by PKSIII Mo29 with malonyl-CoA and palmitoyl-CoA as starter, Caco-2 cells showed significantly reduced cell viability compared to the controls when monitoring LDH (Lactate DesHydrogenase) released (Figure 6). Indeed, an increase of LDH leakage was observed for the extract-exposure cells (*p* < 0.05) indicating cytotoxic effects. Another tumoral cell line was used with the same molecules to see if cells response is specific.

THP1 cells are employed because these cells are immunological targets. Positive results were also observed when THP1 cells were exposed to the same molecules (All molecules are concentrated at 0.012 g/L) (Figure 7). With these cells, we studied mitochondrial activity using MTS bioassay (bromure de 3-(4,5-dimethylthiazol-2-yl)-2,5-diphenyl tetrazolium). In this experiment, mitochondrial activity was significantly reduced after 24 h exposure to the compound with a molecular formula of C_20_H_34_O_3_ (*m*/*z* 321.2400) produced by PKSIII Mo29 with malonyl-CoA and palmitoyl-CoA (as starter) with 11% of affected fraction.

## 3. Discussion

Genes encoding type III Polyketides synthases are found in Dikarya such as *Aspergillus niger*, *Aspergillus oryzae*, *Neurospora crassa*, *Botrytis cinerea* or *Sordaria macrospora* [37,38,39,53,55,57], and recent genomic studies by Navarro-Munoz and Collemare revealed that there are 522 genes encoding these enzymes among 1193 fungal genomes [43]. However, currently, very few fungal pyrone synthase enzymes have been described biochemically. Indeed, there are only 11 PKSIII enzymes from fungi that have been functionally characterized and these all belong to fungal species of terrestrial origin. In this work, we have described the first biochemical characterization of PKSIII from marine fungi, a yeast, *Naganishia uzbekistanensis* strain Mo29 (UBOCC-A-208024) (formerly named as *Cryptococcus* sp.).

Marine fungi were first described in the 1980s and 1990s [9,62]. Thanks to numerous oceanographic cruises, we now know that it is possible to find many fungal species in the marine environment. Among these, the species found mainly pertained to Dikarya (*Basidiomycota* and *Ascomycota*). These species are found in different places in the marine environment such as the water column, associated with macro-organisms like algae, deep-sea sediments and hydrothermal vents [63]. The *N. uzbekistanensis* strain Mo29 (UBOCC-A-208024) was discovered during the MoMARDREAM-Naut oceanographic cruise (2007), on the exploration of the hydrothermal sources of the Rainbow site (-2300 m, Mid-Atlantic Ridge) [14]. Genome sequencing of this fungal species led to identify a gene encoding a PKSIII [45]. Phylogenetic analysis of the PKSIII protein sequence with enzymes from plants and bacteria, revealed that all fungal PKSIII enzymes (619 protein sequences) were grouped into a single branch. The PKSIII enzyme from *N. uzbekistanensis* strain Mo29 strain (UBOCC-A-208024) was not included in the fungal group/cluster. Instead, it clustered with 3 other protein sequences identified in another strain of *Naganishia* sp. and two other strains of *Cryptococcus* sp. [43]. This suggests that sequences of these three PKSIII proteins have followed an independent and early separation in the evolution compared to the other fungal sequences of PKSIII included in this study.

Interestingly, one of the features of these four enzymes is the presence of an additional 74 amino acids extension at the C-terminus. Previous studies in *Sordaria macrospora* and *Botrytis cinerea* have revealed that PKSIII may have a C-terminus extension [53,55] but the contribution of this region to the enzymatic function in these organisms is not known. Here, by expressing forms of the PKSIII Mo29 protein with and without this C-terminus extension, we show that it is not required for the enzymatic activity. Both forms are functional and produce mainly pyrones and some alkylresorcinols.

Analysis of the predicted structure of PKSIII Mo29 identified three catalytic residues: cysteine at position 171 (171C), Histidine at position 352 (352H) and Arginine at position 385 (385N). The next step is to mutate one or the three catalytic residues. This experiment could prove that these residues are involved in the enzymatic activity.

Previous studies of PKSIII have shown that another essential structure for the PKSIII protein is the “tunnel” which allows the entry of different starters such as lauroyl-CoA or stearoyl-CoA. These starters will either be cyclized or branched to the cycle synthesized by malonyl-CoA in the tunnel structure. The majority of the amino acids involved in the tunnel in PKSIII Mo29 protein are identical or similar to those found in the structure of PKSIII from *N. crassa*, from *A. oryzae* and *B. cinerea* (Table 1 and Appendix A) [38,39,41,47,52,53]. Based on the structure of the PKSIII of *N. crassa* (3E1H) and the models of *B. cinerea* and *A. niger*, some amino acids explaining structural differences could be found. In position 144 of the PKSIII from *N. uzbekistanensis* strain Mo29, there is a tyrosine (144Y) whereas in *N. crassa* and the two other fungal proteins, there are two asparagines (PKSIIINc 125N and BcPKS 138N) and an alanine (AnPKS 140A) (amino acids not having the same chemical characteristics). Similarly, at position 212, there is a cysteine (212C), while in the three other fungal sequences, there is a non-polar amino acid (PKSIIINc 190V, BcPKS 203V and AnPKS 205L). In position 236, the PKSIII Mo29 has a serine (that is a polar amino acid) whereas in the three other fungi, there is a non-polar amino acid (PKSIIINc 206G, BcPKS 219G and AnPKS 221A). The other notable difference is in the amino acid that could bind to stearoyl-CoA. For the PKSIII Mo29, at position 355, there is a serine whereas in the other proteins, a nonpolar (309A), an aromatic (321A) or a positively charged amino acid (320R) are found, respectively, in PKSIIINc, BcPKS and AnPKS. These different amino acid changes in the structure of the PKSIII Mo29 protein could explain the fact that it does not synthetize short chain pyrones like some PKSIII proteins characterized biochemically in *A. oryzae* (AnCsyA and AnCsyB) [39,40].

We overexpressed the entire PKSIII Mo29 protein as well as a truncated form lacking the additional 74 aa at the C-terminus. These two proteins forms are dimeric and are active. Indeed, the 74 aa extension in C-terminus does not seem to be involved in the enzymatic activity as in *B. cinerea* and in *S. macrospora* [53,55]. The two proteins are able to produce compounds in the presence of malonyl-CoA but always in the presence of an acyl-CoA starter with a carbon chain longer than 6 units. PKSIII Mo29 does not incorporate small carbon chains (acetyl-CoA, malonyl-CoA (only) and hexanoyl-CoA). No molecules are synthesized in the presence of these three starters. The known fungal pyrone synthases catalyse reactions starting from the acyl-CoA chain and ending with aldol cyclization and/or lactonization [25]. The fungal PKSIII that have been experimentally characterized are divided into two functional groups. The first group uses only long acyl-CoA chains with several malonyl-CoA. We find in this group enzymes from *N. crassa*, *A. niger*, *B. cinerea* and *S. macrospora* [47,53,55]. The second group consists of two enzymes found in *A. oryzae* [38,39,40]. These two enzymes can use starter molecules with chains of different lengths to produce triketide, tetraketide pyrones and alkylresorcinols. These results show that the Mo29 enzyme belongs to the first group (like the enzymes of *N. crassa*, *A. niger*, *B. cinerea* and *S. macrospora*). PKSIII Mo29 produces triketide and tetraketide pyrones in the presence of starter such as octanoyl-CoA, decanoyl-CoA, lauroyl-CoA and palmytoyl-CoA (Table 2). Another study allowed us to highlight that this protein could accommodate very long chain fatty acyl-CoA starter units produced by *E. coli*. Indeed, molecules synthesized by proteins overexpressed by *E. coli* are secreted into the culture medium. Analysis of molecules found in the culture medium show that the two types of proteins (Std and Red) are able to synthesis several molecules having 14 carbons (C14) to 28 carbons (C28). However, the majority of molecules possess 18 and 20 to 24 carbons. Among these molecules are pyrones and alkylresorcinols. The most synthesized molecules are C_22_H_36_O_3_ (*m*/*z* 348.2663) and C_20_H_32_O_3_ (*m*/*z* 320.2349) which are pyrones (triketide pyrones) and C_23_H_38_O_2_ (*m*/*z* 346.2869) (alkylresorcinol) (Figure 5).

The PKSIII Mo29 Std and PKSIII Mo29 Red synthesize the same molecules, pyrones and alkylresorcinols. However, the truncated form seems to produce less product than the entire recombinant protein. These results were observed in In vitro experiments and in *E. coli* culture medium. In the *E. coli* culture medium, all molecules are less produced by PKSIII Mo29 Red than PKSIII Mo29 Std. These results could be explained by the absence of the C-terminus 74 aa extension. The extension could be implicated in the dimer stability or substrates binding. In this study, we have tested only classical substrates and starters such as palmitoyl-CoA; however, we did not identify natural substrates in *N. uzbekistanensis* strain Mo29 strain (UBOCC-A-208024) yet. So, the extension could be playing a role in the entrance of other substrates than those used in vitro.

Several α-pyrones produced by marine and terrestrial fungi have shown cytotoxic activities against several cancer cell lines [58,59,60,61]. The different molecules produced by PKSIII Mo29 In vitro have been brought into contact with different cancer cell lines, such as Caco-2 and THP1 (colon cancer cell lines and leukemic cell lines). Molecule produced by PKSIII Mo29 in the presence of malonyl-CoA and palmitoyl-CoA is a α-pyrone with a molecular formula of C_20_H_34_O_3_ (*m*/*z* 321.2507). When the Caco-2 cells and the THP1 are exposed for 48 h in the presence of 0.012 g/L of this molecule, the cell viability decreases. Therefore, it would mean that this α-pyrone has a cytotoxic effect on these two tumors cell lines. The molecule produced by PKSIII Mo29 in the presence of malonyl-CoA and lauroyl-CoA In vitro is also a triketide pyrone with a molecular formula of C_16_H_26_O_3_ (*m*/*z* 266.1884). THP1 cells were incubated for 48 h with 0.012 g/L of this molecule. Cell viability was reduced by 5%. These two molecules seem to have cytotoxic effects on cancer cell lines like Caco-2 and THP1.

## 4. Materials and Methods

### 4.1. Fungal Strains

The yeast strain studied in these experiments was a *Naganishia uzbekistanensis* strain Mo29 (UBOCC-A-208024) isolated from hydrothermal vents of the Rainbow site (−2300 m, Mid-Atlantic Ridge) during the oceanographic cruise MoMARDREAM-Naut (2007) [14,45]. Yeast was cultivated at 30 °C on culture medium composed of 2% malt extract, 0.3% yeast extract and 1.5% agar during 7 days.

### 4.2. Bacterial Strains

*Escherichia coli* SURE 2 Supercompetent Cells [e14(McrA-) Δ(*mcrCB-hsdSMR-mrr*) 171 *end*A1 *gyrA96 thi-1 supE44 relA1 lac recB recJ sbsC umuC*::Tn*5* (Kanr) *uvrC* [F’ *proAB lacIqZΔM15* Tn*10* (Tetr)]] (Agilent Technologies, Santa Clara, CA, USA) was used as a host strain in order to verify the gene sequences. For protein expression, recombinant pQE-80L (Qiagen, Hilden, Deutschland) vectors were transformed into *E. coli* BL21-CodonPlus(DE3)-RIPL [*E. coli* B F^–^
*ompT hsdS*(rB^–^ mB^–^) *dcm*^+^ Tet^r^ gal λ(DE3) *endA* Hte [*argU proL* Camr] [*argU ileY leuW* Strep/Specr]] (Agilent Technologies). Bacteria were cultivated at 37 °C and 18 °C on Luria-Bertani solid medium (0.1% peptone, 0.05% yeast extract, 0.1% sodium chloride, 0.1% agar) or Luria-Bertani liquid medium (0.1% peptone, 0.05% yeast extract, 0.1% sodium chloride).

### 4.3. Phylogenetic Analysis

The selected set of PKSIII was constituted of 755 aa sequences (Appendix A). Fasta formatted sequences were inserted into the NGPhylogeny.fr pipeline “A la carte” (https://ngphylogeny.fr/) and were analysed as followed. The protein sequences were first aligned using MAFFT then cleaned with the tool trimAl resulting in the selection of 271 informative positions over the 2957 aligned positions. Maximum likelihood phylogenetic analysis according to the best substitution model selection (LG +G + I) was carried out using the PhyML-SMS tool under the standard conditions. The bootstrap analysis of 100 replicates was used to provide estimates for the phylogenetic tree topology and it resulted in a newick file formatted with the program MEGA v10.1.1 to obtain the corresponding simplified tree figure.

### 4.4. Cloning of Polyketide Synthase and Sequence Validation

Total RNA was extracted from *N. uzbekistanensis* strain Mo29 (UBOCC-A-208024) grown on culture medium agar using the RNeasy kit for Plant and Fungi (Qiagen, Hilden, Deutschland ) and cDNAs were synthesised using the GoScript™ Reverse Transcription System kit (Promega, Madison, WI, USA). Then, cDNA coding for *type III pks* gene were amplified by PCR using specific primers. One amplification corresponds to the complete sequence of the gene encoding the entire protein (Standard PKSIII (Std)) whereas the other corresponds to the sequence truncated by the last 74 amino acids, encoding a reduced protein (truncated/reduced PKSIII (Red)). Amplification of the *standard pksIII* transcript was performed with primers PKSIII_Mo29_Forward (5′-CATGCGAGCTCGGTACCGCATCTTCTTCCCATATCCTTGG-3′; *Kpn*I restriction site is underline) and PKSIII_Mo29_StandardReverse (5′-TCAGCTAATTAAGCTTTTACGGTGCCACCTTGTTGTCTCC-3′; *Hind*III restriction site is underlined). PCR was performed using the Phusion^®^ HF DNA Polymerase kit (BioLabs, Ipswich, MA, USA), in 50 µL reaction volumes containing 1 × Phusion GC Buffer, 1 mM of dNTPs mix, 6% of DMSO, 2.5 µM of each primer, 1U of Phusion DNA polymerase and 50 ng of cDNA. The reduced *pksIII* transcript was amplified with primers PKSIII_Mo29_Forward and PKSIII_Mo29_ReducedReverse (5′-TCAGCTAATTAAGCTTTTACTGGATATCGTGACCGCTGGC-3′; *Hind*III restriction site is underlined). PCR was performed using the Pfu DNA Polymerase kit (Promega), in 50 µL reaction volumes containing 1 × Pfu Buffer, 0.4 mM of dNTPs mix, 0.1 µM of each primer, 3U of Pfu DNA polymerase and 50 ng of cDNA. The PCRs were performed on PeqStar 2 × thermocycler (Peqlab). The amplification consisted of an initial denaturation step at 94 °C for 2 min, followed by 30 iterations of 30 s at 94 °C, 30 s at 56 °C, 2 min at 72 °C and a final extension step of 10 min at 72 °C. The amplified fragments fragments were extracted from agarose gels and purified using the NucleoSpin Gel PCR Clean-up kit (Macherey-Nagel, Hoerdt, France), then cloned into pQE-80L (Qiagen) using *Kpn*I and *Hind*III restriction sites and primers PKSIII_Mo29_Forward, PKSIII_Mo29_StandardReverse and PKSIII_Mo29_ReducedReverse. Ligated plasmids were transformed into *E. coli* SURE 2 Supercompetent Cells (Agilent Technologies, Santa Clara, CA, USA) and transformants selected using LB solid medium supplemented with 100 μg/mL of ampicillin. Plasmids were extracted and verified by sequencing (Eurofins Genomics, Luxembourg). A multiple alignment was generated with the two obtained sequences and initial *N. uzbekistanensis* strain Mo29 (UBOCC-A-208024) *pksIII* gene sequence [45], using on-line software MUSCLE (Multiple Sequence Comparison by Log-Expectation) and ClustalW2 program [64]. 

### 4.5. Expression and Purification of Recombinant Proteins

For protein expression, *E. coli* BL21-CodonPlus-RIPL cells (Agilent Technologies, Santa Clara, CA, USA ) transformed with sequence-verified plasmids was grown at 37 °C under agitation in LB liquid medium containing 100 µg/mL of ampicillin until the OD600 nm reached a between 0.4 and 0.6. Isopropyl-β-D-thiogalactopyranoside (IPTG) was added to a final concentration of 1 mM and growth continued for 21 h at 20 °C with agitation. Cells were then collected by centrifugation at 9000 rpm for 15 min and pellets were resuspended in buffer A (50 mM Tris-HCl pH = 7.5, 500 mM NaCl and 50 mM imidazole) supplemented with lysozyme, protease inhibitor mixture and DNase before lysis by sonication. After centrifugation at 14,000 rpm at 4 °C for 30 min, supernatants were passed through 0.2 µm filter. Enzyme purification was performed on an Äkta Avant system at 20 °C (GE Healthcare, Chicago, IL, USA) with protein detection by UV at 280 nm. His-Tagged proteins were purified on immobilized Ni-nickel tetradentate absorbent (NTA) medium, using a HisTrap FF column (GE Healthcare). Proteins were eluted using an isocratic gradient from 0% to 100% of buffer B (50 mM Tris-HCl pH = 7.5, 500 mM NaCl, 500 mM imidazole). In total, 1–2 mL fractions were collected during the gradient. Fractions were concentrated by centrifugation at 4000 rpm and 4 °C for 20 min using 15 mL centricon filters. Proteins were then further purified on a Sephacryl-200 gel filtration column (GE Healthcare), equilibrated in buffer composed of 50 mM of Tris-HCl pH = 7.5 and 200 mM of NaCl. All protein samples were analysed for purity and integrity using 12% SDS-PAGE. Dynamic light scattering analysis was performed using a DynaPro-801 molecular-sizing instrument (Structural Biology platform; SB Roscoff, Roscoff, France) equipped with a microsampler (Protein Solutions, Wyatt Technology, CA, USA). A 50 µL sample was passed through a filtering assembly containing a 0.02 µm filter into a 12 µL chamber quartz cuvette. The data were analysed using the Dynamics 4.0 and DynaLS software (Wyatt Technology, CA, USA).

### 4.6. Enzyme Assays

Experiments were performed by individually testing six different starter acyl-CoAs at 200 μM (acetyl-CoA, hexanoyl-CoA, octanoyl-CoA, decanoyl-CoA, palmytoyl-CoA and lauroyl-CoA) in an assay mixture of 500 μL containing 20 μM malonyl-CoA, 50 μg of purified enzyme, 50 mM Tris HCl, pH 7.5, and 1 mM EDTA (final concentrations). Incubations were performed at room temperature for 2 h and stopped by adding 10 μL of 37% HCl. The products were then extracted with 1 mL of ethyl acetate with acetic acid (100 1).

### 4.7. LC-MS

Recombinant *E. coli* BL21-CodonPlus-RIPL cells were grown at 37 °C under agitation in Luria-Bertani liquid medium containing 100 µg/mL of ampicillin until the optical density at 600 nm reached a value between 0.4 and 0.6. Isopropyl-β-D-thiogalactopyranoside (IPTG) was added to a final concentration of 1 mM and growth continued for 21 h at 20 °C with agitation. For each construct (PKSIII Std and PKSIII Red), identical cultures were grown without addition of IPTG and used as control. 1 mL of cells was collected and centrifuged. Supernatants were collected and polyketide products were extracted twice with a solution of ethyl acetate and acetic acid (100:1). The organic phase was transferred to a glass vial and evaporated under a stream of nitrogen. Extracts were solubilized in acetonitrile RS Grade followed by a 0.22 µm filtration before LC-MS analyses.

The extracts were analysed on an Agilent 6530 Accurate-Mass Q-ToF LC/MS (Agilent Technologies, Santa Clara, CA, USA). Products were separated on a Zorbax Extend-C18 1.8 µm (2.1 × 50 mm) column (Agilent Technologies, Santa Clara, CA, USA) at a flow rate of 300 µL/min. Gradient elution was performed with water (containing 0.1% formic acid and 0.1 mM ammonium formate) and acetonitrile (containing 0.1% formic acid), from 15 to 100% acetonitrile in 25 min. Only negative Electrospray Ionization (ESI-) were used. Online LC-ESI-MS spectral analyses were performed using MassHunter Quantitative Analysis software and Mass Profiler. Identification of the products was performed by direct comparison with the authentic compounds using chemical formula or proposed from accurate *m*/*z* determination and fragmentation patterns. Spectral comparison was performed using MassHunter Mass Profiler software (Agilent Technologies, Santa Clara, CA, USA ).

### 4.8. Cell Culture Conditions

Caco-2 cells, derived from a human colorectal adenocarcinoma, were purchased from ECACC (number 88081201, Salisbury, UK). Cells were cultured at 37 °C and 5% (*v*/*v*) CO_2_ in DMEM supplemented with penicillin (100 UI/mL), streptomycin (100 μg/mL) containing 10% (*v*/*v*) heat-inactivated foetal bovine serum (FBS), 4.5 g/L glucose, 25 mM HEPES, 2% (*v*/*v*), 2 mM of L-glutamine and 1% (*v*/*v*) of non-essential amino acids (NEAA) (Invitrogen, Carlsbad, CA, USA) with weekly passage. Cells between passage 30 and 50 were seeded at a density of 80 × 10^3^ cells/cm^2^ (Sigma–Aldrich, Saint-Louis, MO, USA). In these conditions, cells reached confluence in 3 days and differentiated completely in 21 days.

Human monocytic leukaemia cells (THP-1) were acquired from the European Collection of Cell Cultures (ECACC; number 88081201, Salisbury, UK). THP-1 suspensions were grown in RPMI-1640 medium supplemented with 10% heat-inactivated foetal bovine serum (FBS), 3652 mM L-glutamine, 10,000 Units/mL 1% penicillin 10,000 Units/mL and 10,000 µg/mL 1% streptomycin 10,000 µg/mL (Biochrom GmbH, Berlin, Germany) at 37 °C with 100% relative humidity (RH) in a 5% CO_2_ atmosphere. Cells were grown to a density between 0.2 and 1 × 10^6^ cells/mL as recommended by ECACC. Culture medium was replaced every 3 days with fresh growth medium.

### 4.9. Lactate Dehydrogenase Assays

Caco-2 cells were grown until 21 days post-confluency (differentiated cells) in culture medium in the presence of different PKSIII Mo29 reaction products at concentrations specified. The viability of the Caco-2 monolayers was measured using the CytoTox 96 Non –Radioactive Cytotoxicity Assay (Promega, Madison, WI, USA). After 48 h, 50 µL of cell culture medium was collected from each well and incubated with the substrate solution for 30 min. Absorbance at 490 nm was measured by spectrophotometer. Experiments were repeated three times, six repetitions each. The relative LDH leakage (%) related to control wells containing cell culture medium with DMSO was calculated by [A]test/[A]control × 100. Where [A]test is the absorbance of the test sample and [A]control is the absorbance of the solvent control sample.

### 4.10. Evaluation of Cytotoxicity by Mitochondrial Activity

THP1 cells were plated in 96-well tissue culture plates at a density of 3 × 10^4^ cells/well. The effects on the mitochondrial activity of THP-1 cells exposed to extracts as indicated in the text were studied using the MTS assay kit. Control cultures without extracts but with solvent (DMSO) were included. After 48 h of exposure (acute exposure), the cells were washed with PBS and 20 μL of a freshly prepared MTS/PMS solution was added to the wells. The cells were further incubated for 3 h. The amount of soluble formazan (MTS metabolite) was then quantified by reading the absorbance at 490 nm on a Multiskan FC plate reader (Thermo Scientific, Madison, WI, USA). Cell viability obtained for the negative control (cell cultures not exposed to molecules) was defined as 100% and affected fraction as 0%. Affected fraction mean percentages of three independent experiments ± standard error of the mean (SEM) were used for statistical analyses.

### 4.11. Statistical Analysis for Cell Viability

Statistical analyses were performed using Statgraphics Plus for Windows (version 1.4 StatPoint Technologies, Inc., Warrenton, VA, USA). After verifying the normal distribution of the data and the homogeneity of variances, one-way analyses of variance were used to detect significant differences among means. Least significant differences (LSD) tests were then applied to compare mean values obtained from different culture condition to negative control.

## 5. Conclusions

The PKSIII of *N. uzbekistanensis* strain Mo29 (UBOCC-A-208024) is the first Polyketide Synthase of type III from a marine fungus to be described. This enzyme produces long α-pyrones and alkylresorcinols. These compounds are produced by an enzyme which have some different amino acids changed in the tunnel structure. But, to prove this hypothesis, different amino acid could be replaced by other residues using site-directed mutagenesis and mutated proteins will be tested with short starter like hexanoyl-CoA. In this study, we have discovered that some molecules produced by PKSIII Mo29 have a cytotoxic effect on two tumoral cell lines. At this time, these molecules have no antimicrobial effects on different bacteria (data not shown). We tried to discover molecules produced in *N. uzbekistanensis* strain Mo29 strain (UBOCC-A-208024) by PKSIII. However, at this time, we are unable to determine which natural molecules are synthetized by PKSIII Mo29. Therefore, it is important to continue the exploration of this strain to understand the biosynthetic pathway of polyketides in this marine yeast.

## Figures and Tables

**Figure 1 marinedrugs-18-00637-f001:**
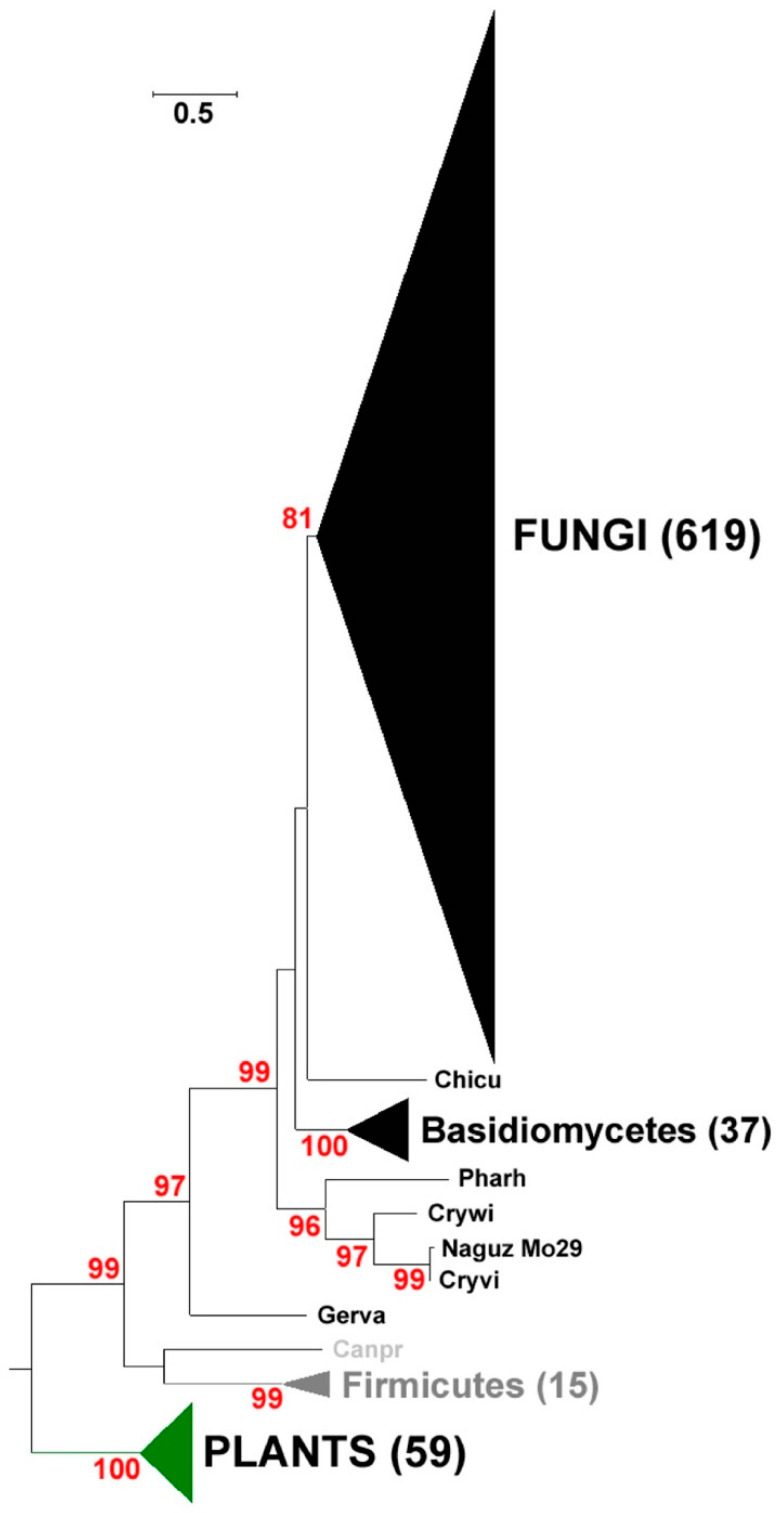
Phylogenenetic representation of putative Type III Polyketide synthase (PKSIII) proteins. The phylogenetic subtree (737 sequences) presented here was constructed using the maximum likelihood approach. Numbers indicate the bootstrap values in the maximum likelihood analysis. Fungal proteins sequences used for this tree were identified by the study of Navarro-Munoz and Collemare [43] and NCBI public database (Appendix A). Published and characterized PKSIII from bacteria and plants are included. Chicu (PKSIII from *Chionosphaera cuniculicola*), Pharh (PKSIII from *Phaffia rhodozyma*), Crywi (PKSIII from *Cryptococcus wieringae*), Naguz Mo29 (PKSIII from *N. uzbekistanensis* strain Mo29), Cryvi (PKSIII from *Cryptococcus vishniacii*), Gerva (PKSIII from *Geranomyces variabilis*) and Canpr (PKSIII from *Candidatus protochlamydia*).

**Figure 2 marinedrugs-18-00637-f002:**
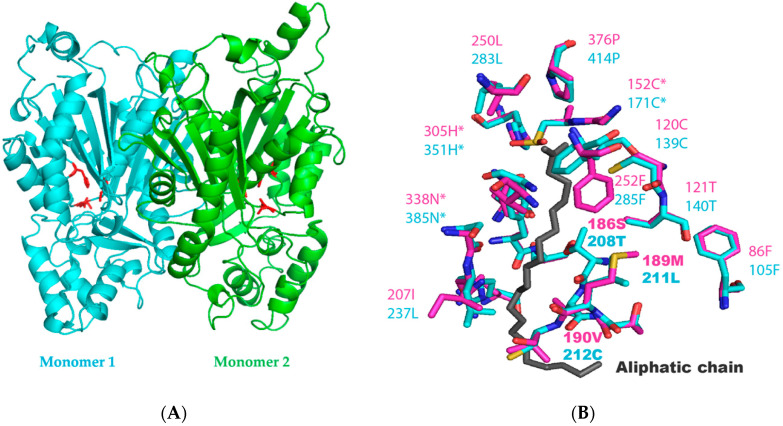
(**A**) Representation of the PKSIII Mo29 model (obtained with SwissModel and PyMOL). The three catalytic residues are shown in red colour in both monomers. (**B**) Comparison of the active site and the long chain substrat-binding tunnel between the structure of PKSIII *N. crassa* (PDB 3E1H) in magenta and the model of PKSIII Mo29 in blue obtained with SwissModel. The comparison was done with PyMOL (The PyMOL Molecular Graphics System, Version 2.0 Schrödinger, LLC.). * Residues of active site. The aliphatic chain (starter) is in grey.

**Figure 3 marinedrugs-18-00637-f003:**
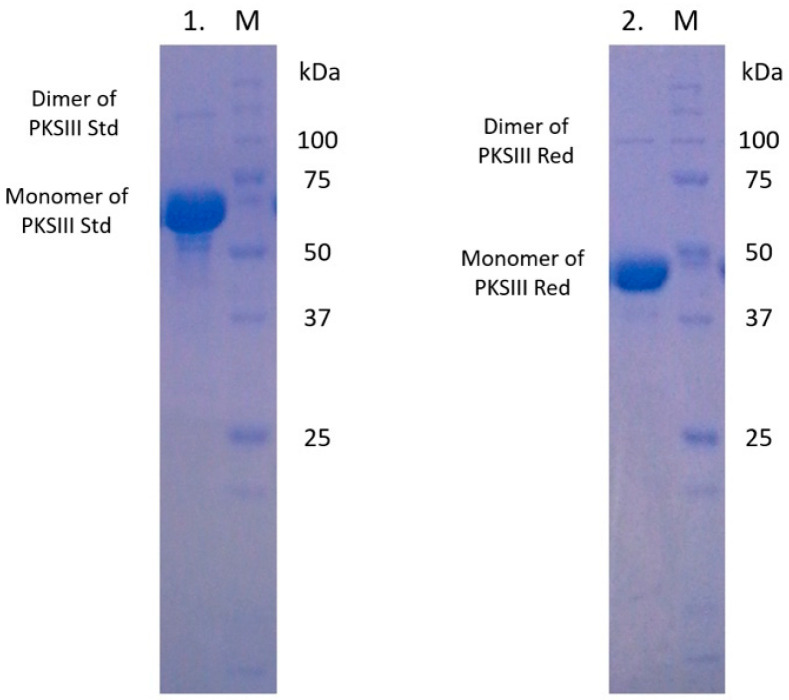
SDS-PAGE of both recombinant PKSIII Mo29 from *N. uzbekistanensis*. Lane 1: pure Std protein after purification by IMAC affinity. M: Molecular weight marker (15–250 kDa). Lane 2: pure Red protein after purification by IMAC affinity. M: Molecular weight marker (15–250 kDa).

**Figure 4 marinedrugs-18-00637-f004:**
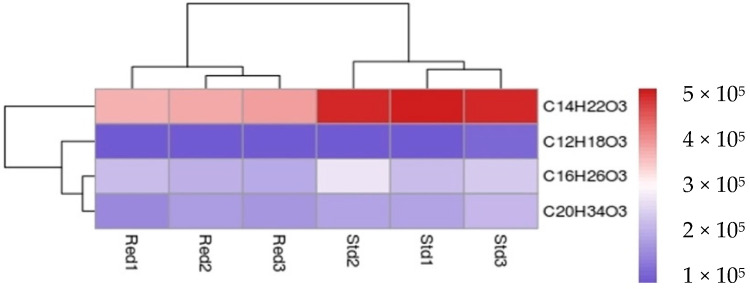
Heatmap. Synthesized molecules abundance by the PKSIII Mo29 Std and PKSIII Mo29 Red during In vitro experiments.

**Figure 5 marinedrugs-18-00637-f005:**
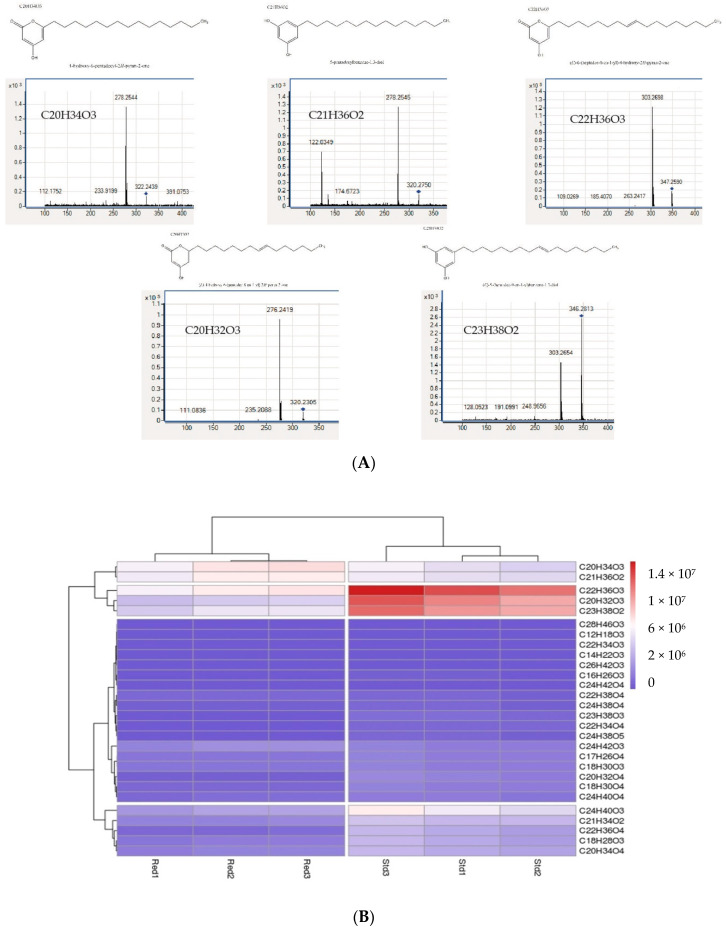
(**A**) Product identification by HPLC-Q-ToFMS. These MS/MS spectra indicate that triketide pyrones and alkylresorcinol are synthesized by PKSIII Mo29 in the presence of acyl-CoA in *E. coli*. (**B**) Heatmap. Synthesized molecules abundance from *pksIII Mo29 std* and *pksIII Mo29 red* expression in *E. coli*. Molecules were clustered based on the *E. coli* supernatants.

**Figure 6 marinedrugs-18-00637-f006:**
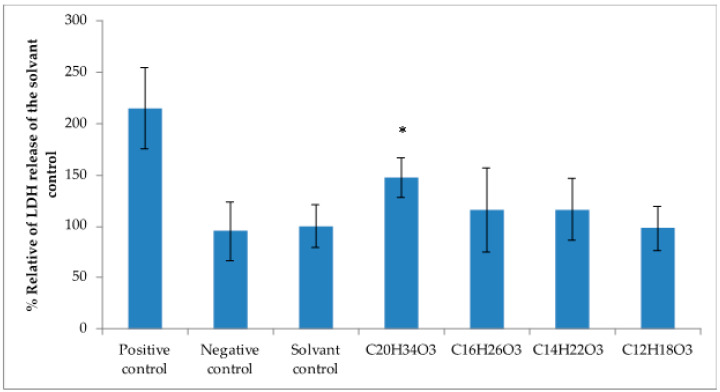
Influence of different culture conditions on Caco-2 cells after 48 h incubation. At the end of the incubation period, the amount of LDH released into the medium was measured. Replicate were done. Standard deviations were established from three biological replicates. * Statistically significant differences at *p* < 0.05 when compared with solvent control. [C_20_H_34_O_3_] = 0.0372 mmol/L, [C_16_H_26_O_3_] = 0.0450 mmol/L, [C_14_H_22_O_3_] = 0.0503 mmol/L, [C_12_H_18_O_3_] = 0.0573 mmol/L.

**Figure 7 marinedrugs-18-00637-f007:**
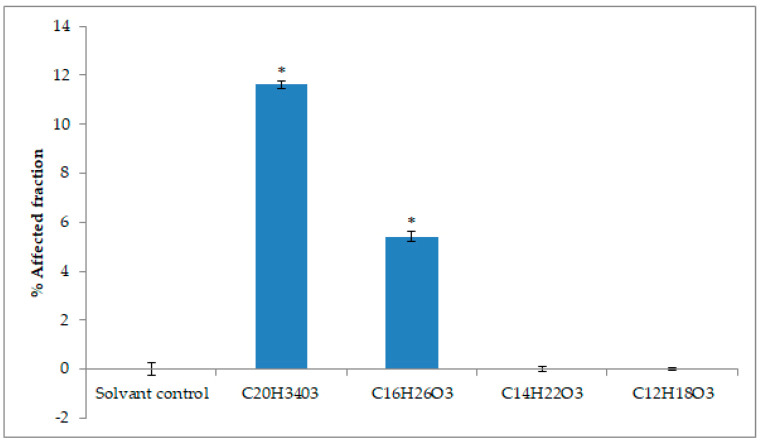
Fraction of THP-1 viability affected ± SEM, quantified using MTS bioassay (N = 3) after 48 h of exposure. * = cell viability measured mean significantly different from solvent control (0%) (*p* < 0.05). [C_20_H_34_O_3_] = 0.0372 mmol/L, [C_16_H_26_O_3_] = 0.0450 mmol/L, [C_14_H_22_O_3_] = 0.0503 mmol/L, [C_12_H_18_O_3_] = 0.0573 mmol/L.

**Table 1 marinedrugs-18-00637-t001:** Comparison of residues lining the substrate-binding region (acyl-CoA starter) in different fungal and bacterial PKSIII proteins.

***Protein***	**Residues Near the Active Site (aa Involved in the Tunnel and Binding with Stearoyl-CoA)**																					
*PKSIIINc*	86F	120C	121T	125N	186S	189M	190V	206G	207I	210F	211S	250L	252F	261V	306P	307G	308G	309A	310T	311I	312L	313S
*PKS18*	109F	143S	144T	148A	205C	210V	211F	220I	221H	224F	225G	264I	266L	275C	314P	315G	316G	317P	318K	319I	320I	321E
*AoCsyA*	101F	135C	136T	140N	201C	204F	205F	221A	222M	225F	226G	266I	268F	277P	323P	324G	325G	326Y	327S	328I	329A	330V
*AoCsyB*	89F	123C	124T	128H	189P	192F	193A	209A	210M	213F	214G	254A	256F	265A	311P	312G	313G	314Y	315A	316V	317L	318V
*BcPKS*	99F	133C	134T	138N	199S	202L	203V	219G	220V	223F	224S	263L	265F	274V	318P	319G	320G	321A	322T	323I	324L	325T
*AnPKS*	101F	135V	136T	140A	201C	204H	205L	221A	222P	225F	226S	265M	267Y	276A	317P	318G	319G	320R	321A	322V	323I	324Q
*PKSIII Mo29*	105F	139C	140T	144Y	208T	211L	212C	236S	237L	240F	241S	283L	285F	294A	352P	353G	354G	355S	356L	357I	358I	359S
***Protein***	**Residues of active site**																					
*PKSIIINc*	152C							305H							338N							
*PKS18*	175C							313H							346N							
*AoCsyA*	167C							322H							355N							
*AoCsyB*	155C							310H							343N							
*BcPKS*	165C							317H							350N							
*AnPKS*	167C							316H							349N							
*PKSIII Mo29*	171C							351H							385N							

**Table 2 marinedrugs-18-00637-t002:** Products synthetized by PKSIII Mo29 Std and PKSIII Mo29 Red in vitro. Molecules were analyzed and detected by HPLC-Q-ToFMS. (ND: Not detected).

Starter Substrat	Products	Rt LC/min	Parent Ion *m*/*z*	Fragment Ion *m*/*z*
*Acetyl-CoA*	ND	/	/	
*Malonyl-CoA*	ND	/	/	
*Hexanoyl-CoA*	ND	/	/	
*Octanoyl-CoA*	C_12_H_18_O_3_	17.50	209.1300	ND
*Decanoyl-CoA*	C_14_H_22_O_3_	19.80	238.1567	194.1599
*Lauroyl-CoA*	C_16_H_26_O_3_	22.05	266.1884	222.1952
*Palmytoyl-CoA*	C_20_H_34_O_3_	26.03	322.2507	278.2554

**Table 3 marinedrugs-18-00637-t003:** Products synthetized by *E. coli* containing PKSIII Mo29 Std and PKSIII Mo29 Red. Molecules were analysed and detected by HPLC-Q-ToFMS. ND: Not detected.

Products	Name of Products	RT (min)	Parent Ion *m*/*z* (ESI(-))	Fragment Ion *m*/*z*
*C_12_H_18_O_3_*	Pyrone	17.5	209.1300	ND
*C_14_H_22_O_3_*	Pyrone	19.735	238.1567	194.1599
*C_16_H_26_O_3_*	Pyrone	21.973	266.1884	222.1952
*C_17_H_26_O_4_*	Pyrone	24.11	293.1800	ND
*C_18_H_28_O_3_*	Pyrone	22.48	292.2038	248.2109
*C_18_H_30_O_3_*	Pyrone	24.125	294.2192	250.2236
*C_18_H_30_O_4_*	Pyrone	20.656	310.2143	125.0231
*C_18_H_32_O_3_*	Pyrone	24.6	295.2200	ND
*C_20_H_32_O_3_*	Pyrone	24.421	320.2349	276.2421
*C_20_H_32_O_4_*	Pyrone	21.27	336.2296	125.0225
*C_20_H_34_O_3_*	Pyrone	26.10	322.2507	278.2554
*C_20_H_34_O_4_*	Pyrone	22.8	338.2450	125.0221
*C_21_H_34_O_2_*	Resorcinol	25.037	318.2555	122.0376 276.2415
*C_21_H_36_O_2_*	Resorcinol	26.57	320.2714	122.0349 278.2545
*C_22_H_34_O_4_*	Pyrone	26.5 (27.2)	361.2400	ND
*C_22_H_36_O_3_*	Pyrone	26.22	348.2663	303.2700
*C_22_H_36_O_4_*	Pyrone	23.21	364.2620	125.0223
*C_22_H_38_O_3_*	Pyrone	27.97	350.2832	305.2855
*C_22_H_38_O_4_*	Pyrone	26.572	366.2766	319.2635
*C_23_H_38_O_2_*	Resorcinol	26.65	346.2869	122.0338 304.2729
*C_23_H_38_O_3_*	Resorcinol	23.455 (27.24)	362.2814	123.0448
*C_24_H_38_O_4_*	Pyrone	25.49	389.2700	ND
*C_24_H_38_O_5_*	Pyrone	22.98	406.2716	125.0232
*C_24_H_40_O_3_*	Pyrone	27.95	376.2975	331.2974
*C_24_H_40_O_4_*	Pyrone	26.657	392.2923	345.2814
*C_24_H_42_O_3_*	Pyrone	30.315	378.3131	333.3145
*C_24_H_42_O_4_*	Pyrone	28.308	394.3075	347.2919
*C_26_H_42_O_3_*	Pyrone	27.279	402.3130	123.0795 357.3123
*C_28_H_46_O_3_*	Pyrone	28.918	430.3442	385.3474

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
