# Peer review of "Identification and Characterization of a New Type III Polyketide Synthase from a Marine Yeast, Naganishia uzbekistanensis"

_marinedrugs, 2020, doi:10.3390/md18120637_

Round 1

Reviewer 1 Report

  • The present work is very interesting as it describes the potential identification of a Type III Polyketide synthase in a marine yeast. However, I have some suggestions for the authors to improve their paper.

    • in title, uzbekistanensis should not appear with the U in capital
    • English form should be revised (e.g. line 32 - these two recombinant PKSIII protein - "protein" should sound "proteins")
    • the scientific name of all species (eg. N. uzbekistanensis line 118) should appear always in italics in the whole text
    • MTS assay measure the cell growth. I suggest to perform a trypan exclusion test to clearly check if the reduction of cell growth is due to toxicity of just cell cycle arrest.
    • Fig 6 and 7 has asterisks not correctly positioned
    • I would like to specifically understand if the phylogenetic tree was based on the whole DNA sequence of PKSIII from the different organisms or only on the DNA sequence corresponding to their respective active site. The sequence you used for the contruction of the phylogenetic tree are also present and registered on scientific databases? Please, report the accession number for each sequence.
    • Did you registered the sequence of of the putative PKSIII gene that you discovered in a scientific database, for instance, GeneBank? I think that you should do it.
    • Table 1 does not appear homogeneously.. numbers and letters should appear all in one or two lines. Moreover, I do not understand why, for instance, position 340 in PKSIII Nc appears between position 261 and 306. It is not corrected! Similarly for the others.

Reviewer 2 Report

Journal: Marine Drugs

Title: Identification and Characterization of a New Type III Polyketide Synthase from a Marine Yeast, Naganishia uzbekistanensis

Manuscript ID: marinedrugs-1027377 

In this manuscript, identification of a putative Type III Polyketide synthase (PKSIII) encoding genes from the marine yeast Naganishia uzbekistanensis strain Mo29 isolated from deep-sea hydrothermal vents, was carried out. The PKSIII of N. uzbekistanensis strain Mo29 (UBOCC-A-208024) is the first Polyketide Synthase of type III from a marine fungus to be described.

Full-28 length and reduced versions of this PKSIII gene were successfully cloned and overexpressed in a bacterial host, Escherichia coli BL21 (DE3). Functions of proteins produced were investigated.

A novelty of this study resides in the fact that some molecules produced by PKSIII Mo29have a cytotoxic effect on two tumoral cell lines. At present, it is has not be possible to determine which natural molecules are synthetized by PKSIII Mo29.

The manuscript is well written and is informative and represents a very interesting base for future researches.

The manuscript deals with aims and scope of the journal Marine Drugs and, in my opinion, can be accepted for publication in Marine Drugs after Minor revision.
